# The importance of round-robin validation when assessing machine-learning-based vertical extrapolation of wind speeds

Nicola Bodini [1] and Mike Optis [1]

[1]National Renewable Energy Laboratory, Golden, Colorado, USA

**Correspondence:** Nicola Bodini (nicola.bodini@nrel.gov)

**Abstract.**

The extrapolation of wind speeds measured at a meteorological mast to wind turbine rotor heights is a key component in a bankable wind farm energy assessment and a significant source of uncertainty. Industry-standard methods for extrapolation include the power law and logarithmic profile. The emergence of machine-learning applications in wind energy has led to several studies demonstrating substantial improvements in vertical extrapolation accuracy in machine-learning methods over these conventional power law and logarithmic profile methods. In all cases, these studies assess relative model performance at a measurement site where, critically, the machine-learning algorithm requires knowledge of the rotor-height wind speeds in order to train the model. This prior knowledge provides fundamental advantages to the site-specific machine-learning model over the power law and log profile, which, by contrast, are not highly tuned to rotor-height measurements but rather can generalize to any site. Furthermore, there is no practical benefit in applying a machine-learning model at a site where winds at the heights relevant for wind energy production are known; rather, its performance at nearby locations (i.e., across a wind farm site) without rotor-height measurements is of most practical interest. To more fairly and practically compare machine-learning-based extrapolation to standard approaches, we implemented a round-robin extrapolation model comparison, in which a random forest machine-learning model is trained and evaluated at different sites and then compared against the power law and logarithmic profile. We consider 20 months of lidar and sonic anemometer data collected at four sites between 50-100 kilometers apart in the central United States. We find that the random forest outperforms the standard extrapolation approaches, especially when incorporating surface measurements as inputs to include the influence of atmospheric stability. When compared at a single site (the traditional comparison approach), the machine-learning improvement in mean absolute error was 28% and 23% over the power law and logarithmic profile, respectively. Using the round-robin approach proposed here, this improvement drops to 20% and 14%, respectively. These latter values better represent practical model performance, and we conclude that round-robin validation should be the standard for machine-learning-based wind-speed extrapolation methods.

## 1   Introduction

Both the preconstruction and operational phases of wind farm projects require an accurate assessment of the wind resource
at the heights of the rotor swept area to forecast generated power (Brower, 2012). With the constant increase of the size of
commercial wind turbines, the direct measurement of wind speed at heights relevant for wind energy production is becoming
more and more challenging because installing tall meteorological masts requires significant costs. Acquiring and deploying
remote-sensing instruments, such as wind Doppler lidars, also involve substantial economic and technical investments. There-
fore, it is common practice to obtain the characterization of the wind resource at the desired heights by vertically extrapolating
the wind measurements available at lower levels (Landberg, 2015).

One of the most widely used methods to extrapolate wind speed from the measurement height to turbine rotor heights is
by using a power law (Peterson and Hennessey Jr, 1978). Despite not having a physical basis in the theory of meteorology,
this simple relationship can provide agreement with measured wind profiles, especially on monthly or annual timescales, thus
justifying its popularity in the wind energy industry. A second commonly used relationship to represent wind profiles is based
on a logarithmic law, more firmly based on the Monin-Obukhov Similarity Theory (MOST, Monin and Obukhov (1954)).
While both these techniques allow for a simple and to a given extent adequate representation of wind profiles, the limits in their
accuracy, especially under conditions of stable stratification, have been shown in various studies (Lubitz, 2009; Optis et al.,
2016). Both stable stratification and wind flow in complex terrain violate the homogeneity assumption of the MOST theory,
thus often deviating from a logarithmic profile and from the empirical power law profile (Ray et al., 2006). Moreover, neither
law is capable of representing specific phenomena that typically occur in the nocturnal stable boundary layer in some regions,
such as low-level jets (Sisterson et al., 1983), whose strong winds are of great benefit for wind energy production (Cosack
et al., 2007). Offshore wind profiles have also been shown to significantly deviate from power law and logarithmic profiles
(Högström et al., 2006).

Significant research has been conducted to overcome the limitations of the conventional methods used to vertically ex-
trapolate the wind resource (Emeis, 2012; Optis et al., 2014; Badger et al., 2016; Optis and Monahan, 2017). More recently,
machine-learning techniques have been applied to explore their potential in predicting wind speed aloft. Türkan et al. (2016)
compared the performance of seven machine-learning algorithms in extrapolating the wind resource from 10 m to 30 m above
ground level (AGL) at a wind farm in Turkey. Mohandes and Rehman (2018) applied deep neural networks to predict wind
speed up to 120 m AGL using lidar measurements in a flat terrain site in Saudi Arabia. Finally, Vassallo et al. (in review) tested
the performance of deep neural networks in extrapolating wind speed as a function of different input features, both in complex
terrain and offshore, using lidar data. In all cases, the machine-learning models are compared against traditional extrapolation
techniques like the power or logarithmic law, and considerable improvements in extrapolation accuracy using machine-learning
techniques have generally been found.

However, these recent studies assess machine-learning model performance at the site at which the model is trained, an approach that we believe is fundamentally biased. During the model training phase, machine-learning models benefit from having knowledge of the rotor-height wind speeds and are therefore highly tuned to the site at which they are trained. By contrast, conventional extrapolation approaches do not have nor require knowledge of rotor-height wind speeds and therefore can generalize to any location where measurements are available at a single level near the surface (for the logarithmic law) or at two levels in the lower part of the boundary layer (for the power law). Furthermore, the evaluation of machine-learning model performance at the site at which it is trained is not practical: if winds at the heights relevant for wind energy production are already known and measured, there is no need for an extrapolation.

To more fairly and practically validate machine-learning-based vertical extrapolation of wind speeds against conventional methods, a "round-robin" approach should be used. Such an approach involves training the model at a given site and then assessing its performance at other sites where rotor-height wind speeds are unknown to the model. This approach would provide a more meaningful and fair comparison against conventional extrapolation methods and would more accurately quantify the advantage of machine-learning-based approaches. To our knowledge, however, no such round-robin validation has been performed in the literature; therefore, the improved performance of machine-learning algorithms over conventional extrapolation methods might currently be overestimated.

In this study, we implement a round-robin validation approach to assess the performance of machine-learning-based vertical extrapolation of wind speeds against conventional methods. Specifically, we contrast a random forest machine-learning algorithm against the power law and logarithmic law. We consider four measurement sites in the central United States located within 50-100 km of each other for the round-robin validation. In Section 2, we describe the lidar and surface measurements used in our analysis. Details on the extrapolation techniques are presented in Section 3. In Section 4, we apply a round-robin approach to test how the predictive performance of the random forest varies with distance, when the learning algorithm is used to predict wind speed at a location different from the training site, and contrast relative performance when implementing a round-robin comparison versus a single-site comparison. We also compare the predictive performance of machine learning with the power law and logarithmic profile. Finally, we analyze how the error in wind-speed vertical extrapolation by the learning algorithm varies with different input features and with height of predicted wind speed. We conclude and suggest future work in Section 5.

## 2   Data: The Southern Great Plains (SGP) Atmospheric Observatory

We use observations collected at the Southern Great Plains (SGP) atmospheric observatory, a field measurement site in north-central Oklahoma, managed by the Atmospheric Radiation Measurement (ARM) Research Facility. To assess the variability in space of the performance of machine-learning-based wind speed vertical extrapolation, we focus on four different locations at the site (Figure 1), over a region about 100 km wide. The site is primarily flat, and its land use is characterized by cattle pasture and wheat fields. Winds mostly flow from the South, with more variability observed in the winter. For our analysis, we use 30-minute average data from 13 November 2017 to 23 July 2019 (for a total of over 29,000 timestamps).

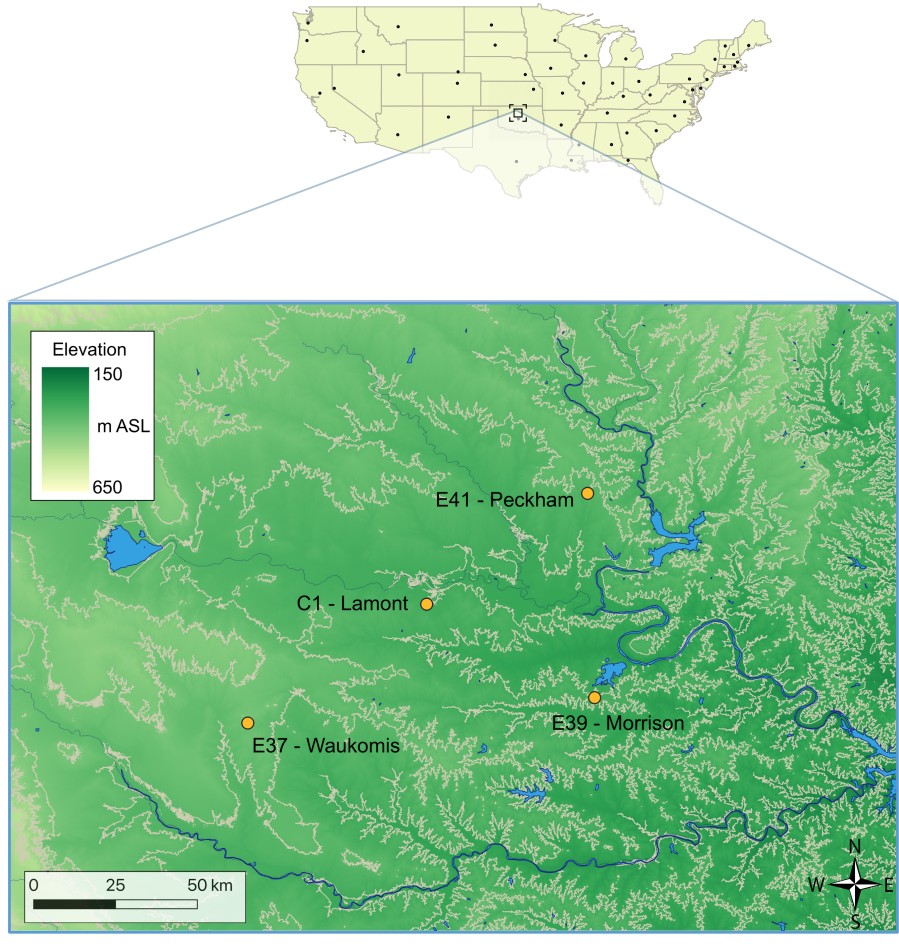

**Figure 1.** Map of the four sites at the Southern Great Plains atmospheric observatory considered in this study. Contour lines at 50-meter intervals are shown in the map. Digital Elevation Model data courtesy of the U.S. Geological Survey.

## 2.1 Lidars

At each of the four locations considered in our study, a Halo Streamline lidar (main technical specifications in Table 1) was deployed. A preliminary intercomparison study of the lidars performed by Atmospheric Radiation Measurement (ARM) research confirmed that all the lidars produce consistent measurements, with correlation coefficients greater than 0.9, and precision less than 0.1 m/s (Newsom, 2012). The lidars performed a variety of scan strategies. For this analysis, we retrieved horizontal wind speed from the full 360° conical scans, which were performed every ∼10-15 minutes and took about 1 minute to complete. We use the velocity-azimuth-display approach in Frehlich et al. (2006) to retrieve the horizontal wind speed from the line-of-sight velocity recorded in the scans. To do so, we assume that the horizontal wind field is homogeneous over the scan volume, and that the average vertical velocity is zero (Browning and Wexler, 1968). We discard from the analysis measurements with a

**Table 1.** Main Technical Specifications of the ARM Halo Lidars

| | |
|---|---|
| Wavelength | 1.5 μm |
| Laser pulse width | 150 ns |
| Pulse rate | 15 kHz |
| Pulses averaged | 20,000 |
| Points per range gate | 10 |
| Range-gate resolution | 30 m |
| Minimum range gate | 15 m |
| Number of range gates | 200 |

signal-to-noise ratio lower than $-21\,\mathrm{dB}$ or higher than $+5\,\mathrm{dB}$ (to filter out fog events), along with periods of precipitation, as recorded by a disdrometer at the C1 site. Finally, processed data were averaged over 30-minute periods. For this study, data from five range gates are used, corresponding to heights of 65, 91, 117, 143, and 169 m AGL. Data recorded at two lowest heights (13 and 39 m AGL) could not be used because of their poor quality, as they lie in the lidar blind zone.

## 2.2 Surface Measurements

Surface data were collected by sonic anemometers on flux measurement systems and temperature probes, which were deployed at each of the four considered sites. The sonic anemometer measured the three wind components at a 10-Hz resolution; processed data are available as 30-minute averages. We use wind speed at 4 m AGL, and turbulent kinetic energy (TKE) calculated from the variance of the three components of the wind flow as:

$$TKE = \frac{1}{2}(\sigma_u^2 + \sigma_v^2 + \sigma_w^2) \tag{1}$$

Also, at each site we calculate the Obukhov length, $L$, to quantify atmospheric stability:

$$L = -\frac{\overline{T_v} \cdot u_*^3}{k \cdot g \cdot \overline{w'T_v'}} \tag{2}$$

where $k = 0.4$ is the von Kármán constant; $g = 9.81$ m s$^{-2}$ is the gravity acceleration; $T_v$ is the virtual temperature (K); $u_* = (\overline{u'w'}^2 + \overline{v'w'}^2)^{1/4}$ is the friction velocity (m s$^{-1}$); and $\overline{w'T_v'}$ is the kinematic virtual temperature flux (K m s$^{-1}$). A linear correction (Pekour, 2004) has been applied to the flux processing to account for sonic anemometer deficiencies in measuring temperature at sites E37, E39, and E41. For the same reason, at these sites, we use $\overline{T_v}$ from temperature and humidity probes at 2 m AGL. Reynolds decomposition for turbulent fluxes has been applied using a 30-minute averaging period, as commonly chosen for boundary-layer processes (De Franceschi and Zardi, 2003; Babić et al., 2012). We consider stable conditions for $L > 0$m, and unstable conditions for $L < 0$m. Data have been quality-controlled, and precipitation periods were excluded from the analysis to discard inaccurate measurements (Zhang et al., 2016).

## 3   Wind-Speed Extrapolation Techniques

In our analysis, we compare the conventional techniques of power law and logarithmic profile for wind-speed extrapolation with a machine-learning random forest. The standard output or "response" variable in our analysis is the 30-minute average wind speed at 143 m AGL. We acknowledge that the resolution of the data used will have an impact on the magnitude of the error values shown in the analysis (as observations at a higher time resolution would likely cause larger extrapolation errors). However, we do not expect the relative comparison between the different extrapolation techniques and the analysis of the predictor importance to be strongly affected by the resolution of the input features used.

### 3.1   Power Law

The first traditional technique we consider assumes a power law to model the wind vertical profile and extrapolate wind speed, $U$, from a height, $z_1$ to $z_2$:

$$U(z_2) = U(z_1) \left( \frac{z_2}{z_1} \right)^{\alpha} \tag{3}$$

where $\alpha$ is the shear exponent. At each site we calculate a time series of $\alpha$ values by inverting Eq. (3), using data at 4 and 65 m AGL. We then use the power-law profile to extrapolate wind speed measured at 65 m AGL up to 143 m AGL.

### 3.2   Logarithmic Law

The second traditional technique we consider assumes a logarithmic profile (Stull, 2012) for the wind speed, $U$, as a function of height, $z$:

$$U(z) = \frac{u_*}{\kappa} \left[ \ln \left( \frac{z}{z_0} \right) - \Psi_m \left( \frac{z}{L}, \frac{z_0}{L} \right) \right] \tag{4}$$

where $u_*$ is friction velocity, $\kappa = 0.41$ is the von Kármán constant, $z_0$ is the roughness length, $L$ is the Obukhov length, and $\Psi_m$ is a function to include a correction based on atmospheric stability. The roughness length, $z_0$, is usually somewhat arbitrarily chosen based on tabulated values, depending on the land cover at the site of interest. To avoid issues connected to the choice of $z_0$ and the large sensitivity of the logarithmic wind profile to it (Optis et al., 2016), we use the following expression that relates wind speed at two levels, $z_1$ (the height where the wind speed is known) and $z_2$ (the height where extrapolated winds are needed):

$$U(z_2) - U(z_1) = \frac{u_*}{\kappa} \left[ \ln \left( \frac{z_2}{z_1} \right) - \Psi_m \left( \frac{z_2}{L}, \frac{z_1}{L} \right) \right] \tag{5}$$

The stability correction, $\Psi_m$, is calculated from an integral over the vertical dimension between the two considered heights, $z_1$ and $z_2$:

$$\Psi_m \left( \frac{z_2}{L}, \frac{z_1}{L} \right) = \int\limits_{z_1/L}^{z_2/L} \frac{1 - \phi_m(\xi)}{\xi} d\xi \tag{6}$$

where the stability function, $\phi_m$, can be chosen from the different formulations recommended in the literature. For stable conditions, we follow the expression proposed by Beljaars and Holtslag (1991), one of the most commonly used in the wind energy community:

$$\phi_{m,\text{stable}}(\xi) = 1 + a\,\xi + b\,\xi\,(1 + c - d\,\xi)\exp[-d\,\xi] \tag{7}$$

where $a = 1$, $b = 2/3$, $c = 5$, and $d = 0.35$. For unstable conditions, we use the widely accepted formulation by Dyer and Hicks (1970):

$$\phi_{m,\text{unstable}}(\xi) = (1 - 16\,\xi)^{-1/4} \tag{8}$$

## 3.3 Random Forest

The main focus of this study is to contrast the validation of machine-learning-based wind-speed extrapolation using a single-site versus a round-robin approach. Therefore, we defer an exhaustive comparison of different machine-learning algorithms to a later study and only consider a relatively simple random forest in this analysis. A random forest is an ensemble of regression trees, which are trained on different random subsets of the training set. The final prediction is then calculated as the average from the single trees. For the analysis, we used the `RandomForestRegressor` module in Python's Scikit-learn (Pedregosa et al., 2011). Additional details on random forests can be found in machine-learning textbooks (e.g. Hastie et al. (2005)).

The input features used for the wind-speed extrapolation are listed in Table 2. As wind speeds often show a diurnal cycle in response to atmospheric stability (Barthelmie et al., 1996; Zhang and Zheng, 2004), we have included multiple variables to capture the diurnal variability in the atmospheric boundary layer: Obukhov length, TKE, and time of day. To preserve the cyclical nature of time of day (i.e., hour 23 and hour 0 being close to each other), we calculate the sine and cosine[1] of the normalized time of day and include these two input features to represent time in the learning algorithm. We note that when similar techniques are applied to more complex sites, the Obukhov length might not be well-suited to capture atmospheric stability in complex terrain (Fernando et al., 2015), and therefore an accurate choice of the input variables as a function of the specific topography is recommended.

### 3.3.1 Hyperparameter Selection

To create a more accurate algorithm, hyperparameters need to be set before the learning process starts. For the random forest, we consider the hyperparameters listed in Table 3, which also shows the values sampled. We use a five-fold cross validation to evaluate different combinations of the hyperparameters, with 30 sets randomly sampled at each site. We use 80% of the data in the cross-validation, while the remaining 20% (selected without shuffling the original data set to avoid unfair predicting performance improvement because of auto-correlation in the data) is held out for independent testing. The performance of the model is evaluated based on the root-mean-squared error between measured and predicted wind speed at 143 m AGL. The set of hyperparameters that leads to the lowest root-mean-squared error is selected and used to assess the final performance of

---

[1]both needed because each value of sine only (or cosine only) is linked to two different times.

**Table 2.** Input Features Considered in the Analysis for the Random Forest Algorithm

| Input feature | Acronym | Measurement height (m AGL) |
|---|---|---|
| 30-minute average wind speed from lidar at 65 m AGL | WS 65 m | 65 |
| sine of time of the day | time | - |
| cosine of time of the day | | |
| 30-minute average wind speed from sonic anemometer at 4 m AGL | WS 4 m | 4 |
| Turbulent kinetic energy | TKE | 4 |
| Obukhov length | L | 4 |

**Table 3.** Algorithm Hyperparameters Considered for the Random Forest and Their Considered Values in the Cross Validation

| Hyperparameter | Possible Values |
|---|---|
| Number of estimators | 10–800 |
| Maximum depth | 4–40 |
| Maximum number of features | 1–6 |
| Minimum number of samples to split | 2–11 |
| Minimum number of samples for a leaf | 1–15 |

the learning algorithm, described in Section 4. A table with the selected sets of hyperparameters at each site is shown in the Appendix.

## 4 Results

A robust validation of the proposed machine-learning approach for wind-speed vertical extrapolation requires testing the method at sites different from the one used for training. We therefore apply a round-robin approach to train a random forest at each of the four sites, using the input features listed in Table 2, and then test it to extrapolate 30-minute wind-speed data at 143 m AGL at the remaining three sites. Figure 2 shows a heat map of the testing MAE found from this round-robin validation. As expected, the random forest provides the most accurate results when it is tested at the site where it is also trained. For all the considered cases, we find a larger MAE when considering the more practical application of a learning algorithm used to extrapolate winds at a site where it has no knowledge of the winds at the desired height. For all of the considered sites, the MAE increases about 10–15% when the algorithm has no prior knowledge of measured hub-height wind speeds. Different results can be expected when considering sites with a more complex topography, or when performing the round-robin approach over different spatial separations. Moreover, we can expect the performance comparison to be influenced not only by the pure separation between training and testing sites, but also by the different forcings that each specific site experiences. Notably, Bodini and Optis (in review) compared the extrapolation performance of the proposed random forest approach before and after

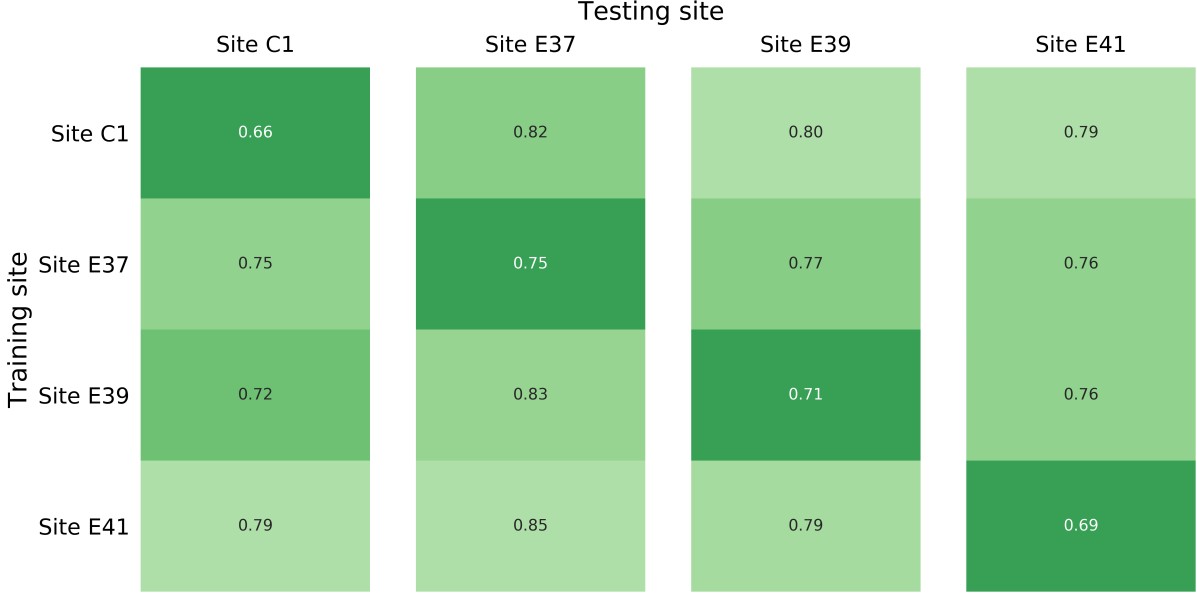

**Figure 2.** Testing mean absolute error (MAE) in predicting 30-minute average wind speed at 143 m AGL for the different sites, as a function of the site used to train the random forest

a wind farm was built in the vicinity of site C1, and found an increase in MAE up to 10% if waked data are not included in the training set. Therefore, to fully exploit the performance of the proposed machine learning approach in extrapolating the wind resource at sites different from the training one it is essential to build a training set of observations which can encompass the specific atmospheric conditions representative of the desired testing site.

The round-robin validation of the machine-learning approach can be completed by comparing the proposed approach with the predictions from conventional techniques for wind-speed vertical extrapolation. In fact, the considered traditional extrapolation laws have a "universal" nature because they can be applied at any site without requiring knowledge of the wind speed at the extrapolation height. Therefore, a fair comparison with the proposed machine-learning approach needs to include a learning algorithm tested at a site where it has no previous knowledge of the wind speed at the desired height. Following the round-robin validation described earlier in this section, we summarize the testing MAE values for all of the approaches we considered in this study, at the four sites, in Figure 3. For the random forest, we include the MAE obtained both when training and testing sites coincide as well as the average results from the round-robin validation. We find that the random-forest approach outperforms the conventional techniques, even when the testing and training sites are different (at the distances sampled in our analysis), although with a reduced decrease in MAE. The percentage reduction in MAE achieved by the random forest over conventional techniques is summarized in Table 4. When evaluated at a single site, we find that the random-forest approach achieves a 23%

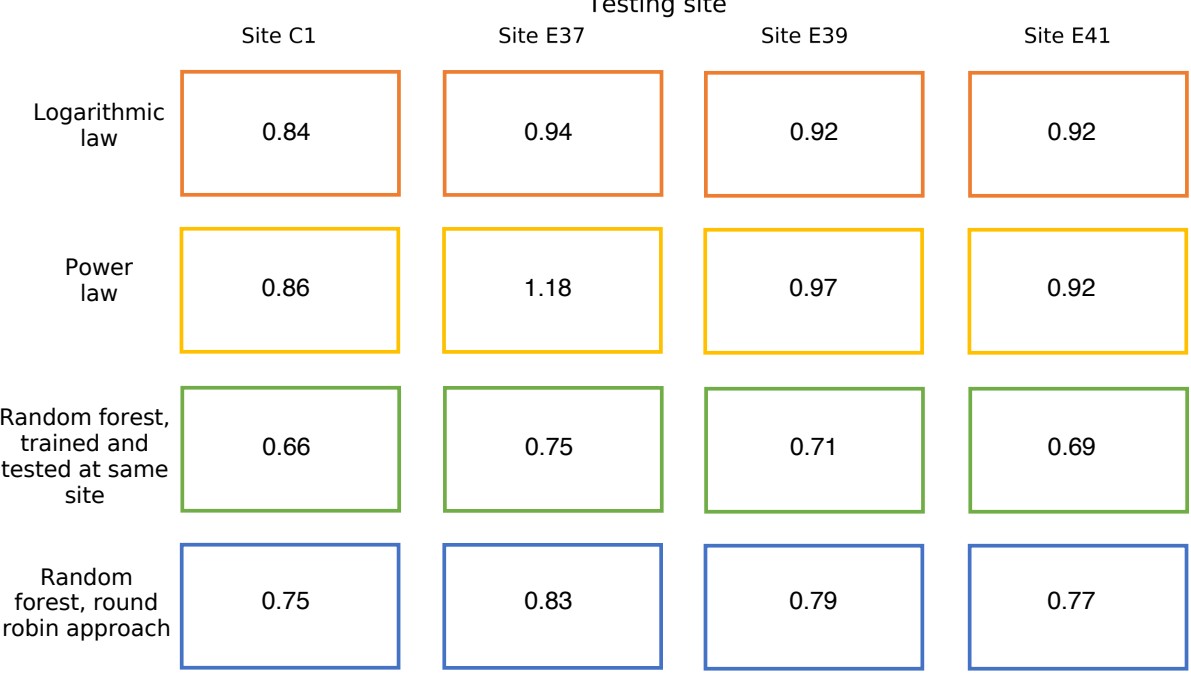

**Figure 3.** Testing MAE in predicting 30-minute average wind speed at 143 m AGL for the different sites, and the different techniques considered in the study

**Table 4.** Percentage Reduction in Wind-Speed Extrapolation MAE from the Random Forest Approach Over the Logarithmic Law and Power Law.

|  | Testing site | | | | Average |
|---|---|---|---|---|---|
|  | C1 | E37 | E39 | E41 |  |
| Error reduction relative to LOGARITHMIC LAW |  |  |  |  |  |
| Learning algorithm trained at the same site | −22% | −21% | −23% | −25% | −23% |
| Learning algorithm trained at a different site | −11% | −12% | −15% | −17% | −14% |
| Error reduction relative to POWER LAW |  |  |  |  |  |
| Learning algorithm trained at the same site | −24% | −36% | −27% | −25% | −28% |
| Learning algorithm trained at a different site | −13% | −30% | −19% | −16% | −20% |

reduction in MAE compared to the logarithmic law, and a 28% reduction with respect to the power law. When the round-robin validation is taken into account, the reduction in MAE decreases to 14% and 20%, respectively.

For the comparison with the power-law predictions, a few additional caveats on the calculation of the wind shear exponent, $\alpha$, are needed. While we acknowledge that determining $\alpha$ using wind-speed data at 4 and 65 m AGL is not ideal and does not realistically reproduce the standard industry approach (where the lower height is typically around 40 m), wind-speed measurements at other heights below 65 m AGL were not available for the considered lidar data set. To assess whether this choice is responsible for the difference in performance between power law and random forest, we calculated a second set of $\alpha$ values by using wind-speed data at 65 m and 91 m AGL, and then extrapolated wind speed from 91 m AGL up to 143 m AGL. We then compared the power-law prediction with the results from a random forest used to predict wind speed at 143 m AGL and trained by adding wind speed at 91 m AGL to the input feature set described in Table 2. We find that the random forest still outperforms the power law, although with a reduced difference in MAE between the two methods (results shown in the Supplement), even under the round-robin approach.

In addition, it is important to check whether the results of the performance comparison are affected by the time resolution at which the shear exponent $\alpha$ is calculated. Wind energy consultants apply a variety of methods to calculate shear (Brower, 2012): one could calculate shear values at each timestamp (as done in our analysis), or use a single average shear exponent, or consider various shear values based on bins of wind direction and/or time of day. To compare the time series-based shear calculation with its most different approach, we test the performance of the power law in extrapolating the average wind resource from 65 m AGL to 143 m AGL using only a single mean value for the shear exponent, calculated as the average of the $\alpha$ values at each considered timestamp. We find that the average extrapolated wind speed from the random forest approach still has a smaller error compared to the average extrapolated wind speed using the mean shear value, at all the considered sites (across-site MAE for random forest is 0.01 m s$^{-1}$, for power law is 0.13 m s$^{-1}$). Given the overall small MAE values found for both methods, we can also conclude that machine-learning-based extrapolation approaches are most beneficial for time series-based extrapolations, as deficiencies in conventional approaches tend to average out more when considering the long-term average results.

To further validate our performance comparison, it is important to assess whether our results hold when wind speed is extrapolated to different heights. To assess this dependence, at each site we tested and trained four random forests using all the input features in Table 2 to predict the 30-minute average wind speed at each of the four heights where measurements from the lidars were available: 91, 117, 143, and 169 m AGL. We then extrapolated wind speeds at the same four levels, using both the power law and the logarithmic profile. Figure 4 shows how the testing $R^2$ and MAE, vary with the height of the target wind speed, as across-site average, for the three considered extrapolation techniques. The predicting performance of all three methods degrades with height; however, the random forest outperforms the conventional techniques at each of the considered levels. Notably, we find that the performance of the random forest degrades more slowly with height than the conventional extrapolation methods, highlighting the limitations of these conventional methods over large vertical extrapolation ranges. As an application of the performance of the random forest in predicting wind speed at higher heights, we present the case study of a LLJ in a companion paper (Bodini and Optis, in review).

Finally, it is important to determine whether the machine-learning-based approach outperforms the conventional techniques in all atmospheric stability conditions, and, if so, in which conditions the proposed approach is more beneficial. To complete

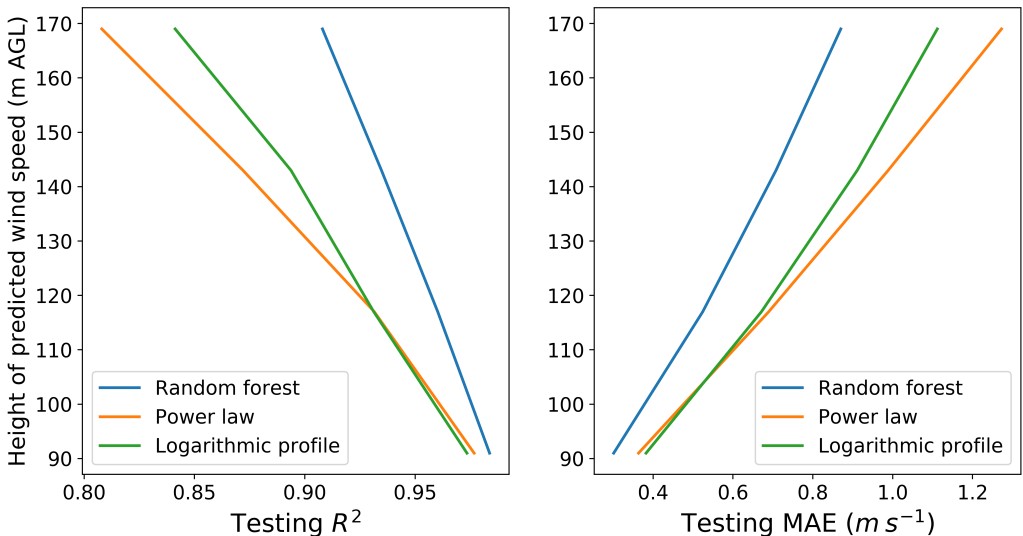

**Figure 4.** Testing $R^2$ and MAE as a function of the height of the extrapolated predicted wind speed, for the three considered techniques

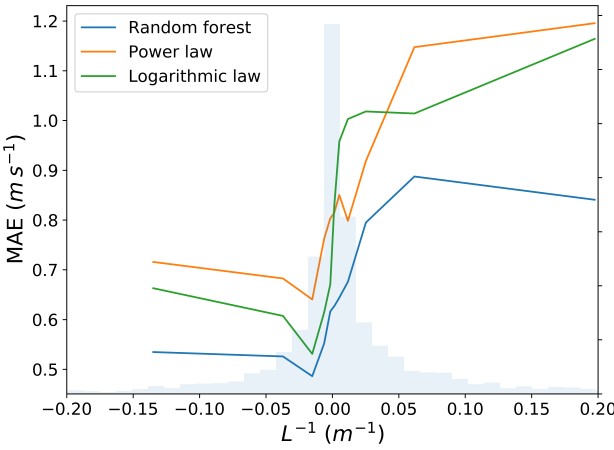

**Figure 5.** Testing MAE in predicting wind speed at 143 m AGL as a function of atmospheric stability, measured in terms of the inverse of the Obukhov length, for the random forest, power law, and logarithmic law, at the C1 site. The distribution of $L^{-1}$ is shown in light blue.

this analysis, we bin the MAE for the three techniques, based on the inverse of the Obukhov length (Figure 5). Data were divided into 12 equally populated groups, based on $L$, and the MAE was calculated for each group and each technique. The random forest shows the lowest error across all considered stability bins. Moreover, we see that the machine-learning-based approach provides the largest reduction in MAE over the conventional techniques under strongly stable conditions.

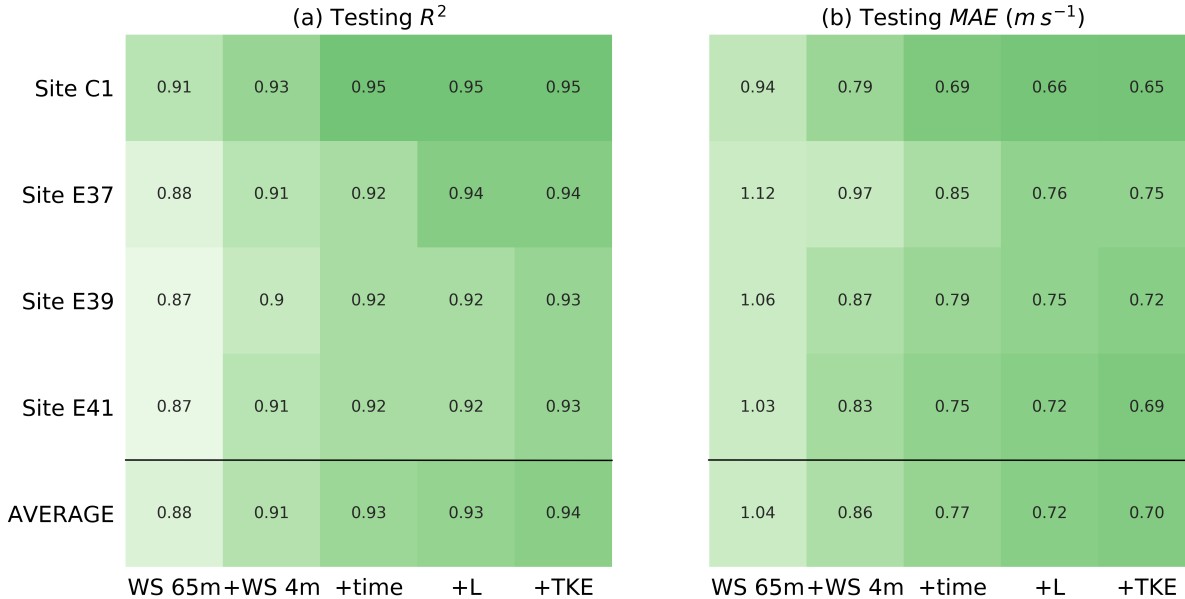

**Figure 6.** Testing $R^2$ and MAE in predicting the 30-minute average wind speed at 143 m AGL for the different sites and input feature combinations

To better understand the strong performance of the random forest in stable conditions, we examine its performance as a function of the set of input features used in the algorithm. Figure 6 shows the testing $R^2$ coefficient and MAE in predicting wind speed at 143 m AGL for different sets of input features at each site and averaged across the four sites. To investigate the potential benefit of including the effects of atmospheric turbulence and stability, we first consider as a base case a random forest that only uses wind speed at 65 m AGL to predict wind speed at 143 m AGL. Then, we progressively add surface winds, time of day (the simplest proxy to include information connected to atmospheric stability), Obukhov length, and finally, TKE. When the random forest is trained using only wind speed at 65 m and 4 m, AGL provides a mean absolute error of $0.86\,\mathrm{m\,s^{-1}}$. Critically, this value is approximately the same magnitude of the power law and logarithmic profile performance. When the time of day, Obukhov length, and TKE are added as input features to the random forest, we find a 20% improvement in the predictive performance, with a further reduction in MAE of 20% ($0.70\,\mathrm{m\,s^{-1}}$ on average). Therefore, the machine-learning-based approach shows improved predictive performance, thanks to its ability to account for atmospheric stability without the need of explicit physical parameterizations, as in the case of the logarithmic profile.

Additional information on the sensitivity of the extrapolated wind speed on the different input features can be provided by considering the partial dependence plots and the predictor performance from the random forest used to predict wind speed at 143 m AGL at site C1 (similar results found at the other sites are not shown). Figure 7 shows the partial dependence plots, which show the marginal effect of each input feature on the predicted extrapolated wind speed (Friedman, 2001). We note that the values on the y-axes have not been normalized, so that large ranges indicate strong dependence of extrapolated wind

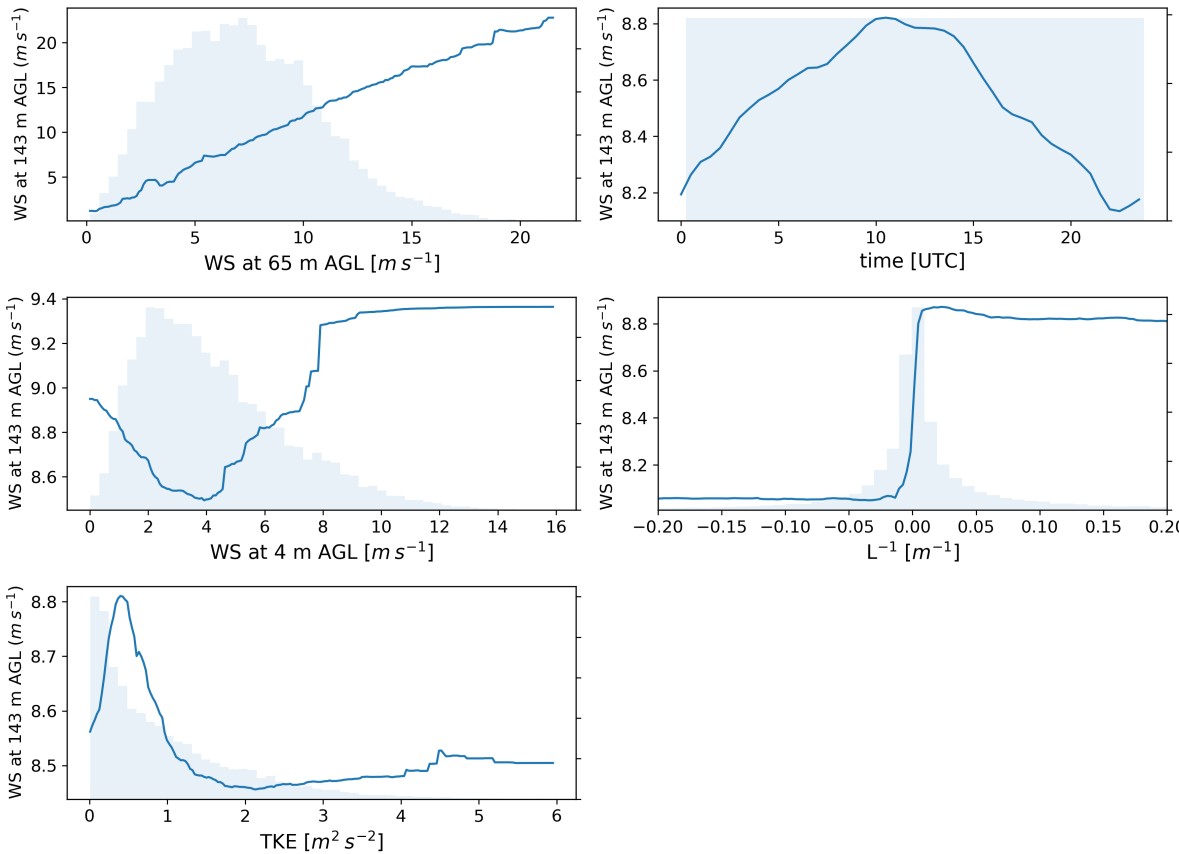

**Figure 7.** Extrapolated wind-speed dependence on individual features for the C1 site. The distribution of each feature is shown in light blue.

speed on the feature, whereas small ranges show weaker dependence. Distributions of the input features are also shown, which help distinguish densely populated regions, with strong statistical relationships, and sparsely populated regions, with weaker statistical relationships. For time of day, the one-dimensional plot shown is derived as a subsample of the two-dimensional partial dependence plot, which was obtained by evaluating the sensitivity of extrapolated wind speed on both the sine and cosine of the normalized time. The key relationships shown in Figure 7 can be summarized as follows:

- Wind speed at 65 m AGL shows a strong positive relationship with extrapolated wind speed at 143 m AGL, with the largest sensitivity among all of the input features, as shown in the plot by the large range of values in wind speed at 143 m AGL.

- Extrapolated wind speed has a clear dependence on time of day, with a distinct diurnal cycle and a peak at approximately 10 UTC (4 a.m. local standard time), and a minimum at 23 UTC (5 p.m. local standard time).

**Table 5.** Predictor importance for the random forest used to extrapolate winds at 143 m AGL at site C1

| Predictor | Relative importance |
|-----------|---------------------|
| WS 65 m | 68% |
| WS 4 m | 18% |
| time | 3% |
| L | 8% |
| TKE | 3% |

- Surface wind speed has a moderate impact on extrapolated wind speed. A minimum in predicted wind speed at 143 m AGL is found for relatively low wind speed at 4 m AGL ($\sim 4\,\mathrm{m\,s^{-1}}$), followed by a systematic increase of extrapolated winds with surface winds. We interpret the negative trend observed for low surface winds as an effect of the fact that very stable conditions are often associated to decoupling, with very low surface wind speeds and increased winds aloft, due to suppressed turbulent mixing.

- Extrapolated winds consistently show, per time of day, a strong relationship with atmospheric stability when quantified by the Obukhov length (whose inverse is shown in the plot to avoid discontinuities). Stable conditions show stronger winds compared to unstable conditions, with a sharp increase under neutral conditions.

- TKE has a smaller impact on extrapolated winds, with a peak for TKE $\sim 0.5\,\mathrm{m^2\,s^{-2}}$ and a subsequent decrease in extrapolated wind speed as TKE increases, again consistent with what we found in terms of atmospheric stability.

The results of the analysis of the predictor performance are listed in Table 5. As already suggested by the partial dependence analysis, wind speed at 65 m AGL is the predictor with the largest importance in extrapolating wind speed at 143 m AGL. However, all the considered surface observations account for over 30% of the overall performance of the random forest. In particular, the addition of the Obukhov length to include direct atmospheric stability information in the algorithm has a not-negligible 8% importance. Overall, the results show the importance of including surface data, especially information connected to atmospheric stability, when vertically extrapolating wind speed, together with the more conventional use of wind-speed aloft.

## 5 Conclusions

Vertically extrapolating wind speeds is often required to obtain a quantitative assessment of the wind resource available at the heights of the rotor swept area of commercial wind turbines. Conventional techniques traditionally used for this purpose, namely a power law and a logarithmic profile, suffer limitations that increase project uncertainty, ultimately leading to increased financial risks for wind energy production. To overcome these drawbacks, machine-learning techniques have been proposed as a novel and alternative approach for wind-speed extrapolation. A fair and practically useful evaluation of the performance of machine-learning-based approaches needs to extrapolate wind speed at a site where the algorithm has no prior knowledge of

the wind speed at the desired height (i.e., at a testing site different than the training one). However, the literature on the topic does not include such validation.

In our analysis, we have performed the first round-robin validation of a random-forest approach to extrapolate wind speed, using 20 months of lidar and sonic anemometer observations from four locations, spanning a 100-km-wide region in the central United States. For the performance of the learning algorithm, we find that including surface atmospheric measurements, and atmospheric stability in particular, reduces the mean absolute error in extrapolated winds by over 30%, compared to including a learning algorithm that only uses wind-speed aloft as input. The benefit of including more physical parameters in a data-driven model clearly demonstrates its importance. Moreover, using a constant set of input features, we find that the accuracy of the random forest decreases as the height of the extrapolated winds increases.

Our proposed approach achieves, on average, a 25%-accuracy improvement over the use of conventional power law and logarithmic profile for wind-speed extrapolation when the algorithm is trained and tested at the same site. This improvement is reduced to 17% when considering the round-robin validation. Therefore, we have confirmed that the random-forest approach outperforms conventional techniques for wind-speed vertical extrapolation, even under a more robust round-robin validation, which we recommend to avoid overestimating the potential performance of machine-learning techniques, which could lead to underestimation of the uncertainty in wind speed estimates. In real world applications, a machine learning algorithm could be trained on observations collected by a single lidar, and then used to extrapolate wind speed at nearby locations, where only much cheaper short meteorological masts would need to be installed.

Future work can expand our round-robin approach by considering different machine-learning algorithms. In addition, the influence of different topographic conditions on the performance of machine-learning-based approaches for wind-speed vertical extrapolation can be considered. Finally, a similar analysis using offshore data could be replicated to help further foster the offshore wind energy industry, specifically the extrapolation of buoy-based, near-surface measurements of wind speed.

*Code and data availability.* Data from the Southern Great Plains atmospheric observatory are publicly available at https://www.arm.gov/ capabilities/observatories/sgp.

## Appendix A: Optimized Hyperparameter Values

Table A1 shows the optimized values of the random forest hyperparameters for each site, as a result of the cross validation.

*Author contributions.* NB performed the analysis on the Southern Great Plains data, in close consultation with MO. NB wrote the manuscript, with significant contributions by MO.

*Competing interests.* The authors declare that they have no conflict of interest.

**Table A1.** Algorithm Hyperparameters Considered for the Random Forest and Their Selected Values for Each Site as a Result of Cross Validation

| Hyperparameter | Possible values | Chosen value | | | |
|---|---|---|---|---|---|
| | | Site C1 | Site E37 | Site E39 | Site E41 |
| Number of estimators | 10 - 800 | 695 | 729 | 614 | 683 |
| Maximum depth | 4 - 40 | 13 | 30 | 15 | 19 |
| Maximum number of features | 1 - 6 | 3 | 3 | 3 | 3 |
| Minimum number of samples to split | 2 - 11 | 3 | 2 | 10 | 8 |
| Minimum number of samples for a leaf | 1 - 15 | 6 | 4 | 1 | 9 |

*Acknowledgements.* Data were obtained from the Atmospheric Radiation Measurement (ARM) Program sponsored by the U.S. Department of Energy, Office of Science, Office of Biological and Environmental Research, Climate and Environmental Sciences Division. We thank Dr. Paytsar Muradyan for giving us insights on how to use ARM data.

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
