# Peer review of "The importance of round-robin validation when assessing machine-learning-based vertical extrapolation of wind speeds"

_Wind Energy Science, 2020_

## Author Comment (AC1) · 24 Jan 2020

2. Data: The Southern Great Plains (SGP) Atmospheric Observatory

We use observations collected at the Southern Great Plains (SGP) atmospheric observatory, a field measurement site in north-central Oklahoma, managed by the Atmospheric Radiation Measurement (ARM) Research Facility. To assess the variability in space of the performance of machine-learning-based wind speed vertical extrapolation, we focus on four different locations at the site (Figure 1), over a region about 100 km wide. The site is primarily flat, and its land use is characterized by cattle pasture and wheat fields. For our analysis, we use data from 13 November 2017 to 23 July

2019.

---

## Referee Comment (RC1) · Dariush Faghani (Referee) · 3 Feb 2020

The paper presents a robust measurement campaign and analysis results. The thoroughness of the various analyses is commendable. The sensitivity analysis at the end of the paper is particularly insightful as it helps shed light on the reasons for the performance of the machine-learning (ML) approach used by the authors. Indeed, the risk when using ML is to blindly depend on a black box which may, or may not, provide reliable output. especially for new situations for which no data was included in the training set.

The following questions and comments are provided in the hope of enhancing the

readability and overall reach of the paper:

0. One could argue that power law and log law are also machine learning approaches - even though they are simple regressions!

1. In the intro, low-level jets (LLJ) are mentioned. Provide some more background as they are not ubiquitously present, nor relevant. Or specify that " in some regions ...".

2. Provide a better presentation of the measurement campaign - notably, do not forget to add the missing paragraph which was posted: - Site description - Typical wind regime description - Lidar precision/accuracy/validation/testing discussion as the wind industry is still considering scanning lidars with a lot of caution. Or, provide discussion that high lidar accuracy is irrelevant in this context because ...  - Provide an idea of the total number of data samples used. - Any data quality applied?

3. The wind industry also uses by-sector and/or by-hour-of-day vertical extrapolation. These are targeting a couple of shortcomings the authors note, namely: stability and terrain complexity. It would be useful to add this in the discussion - or even better, in the analysis.

4. My understanding is that the authors optimized the hyper-parameters by making use of available target-height measurements. So what is the authors' suggestion to fine-tune these parameters in the absence of target-height measurements? Could we contemplate a database of parameters for specific site conditions? Other? More generally, how their round-robin results could be leveraged, used on site?

5. Lines 192 and following: any particular reason comparison results for the specific use case under discussion were not more thoroughly reported?

6. Personally, I find the last sections of the paper to be the most valuable ones! Without suggesting to re-write the whole paper, I submit the following ideas for author's consideration: - Put the emphasis on the fact that more physical parameters where included in a data-driven model, and their impact on model performance was investigated and

fully understood (cf. sensitivities). - The model seems to out-perform standard models, even under round-robin conditions (which is indeed a better way of assessing the model). - The model could be used for a given site as follows (might need more thought to be put here ...)

Thank you for having submitted a paper which makes a balanced and useful use of ML!

---

## Referee Comment (RC2) · Raghavendra Krishnamurthy (Referee) · 4 Feb 2020

The article "The importance of round-robin validation when assessing machine-learning-based vertical extrapolation of wind speeds" by Bodini & Optis details a round robin approach to vertical extrapolation from the 4 ARM SGP Doppler lidars using a random forest algorithm. The paper is well written and the reviewer agrees the necessity of a round-robin type approach to assess the accuracy of machine learning algorithms and make it more universal. Below are some comments/questions which probe into some of the details of the paper and would improve the paper if addressed in the next version.

1. Line 61: Extrapolation is not only generally done up to Hub-height but through the rotor swept area. So, I am not sure I follow the author's argument here, that if hub-height winds are available extrapolation is unnecessary. The approach to go above hub-height can also be treated as a "Gap filling" approach for met-masts (when Lidars are moved around from one location to the other for a short period). Please clarify.

2. For power law type extrapolations, measurements not only at the surface but at multiple heights is needed to estimate the dynamic power law exponent. So please define what you mean by near-surface in the paper? Is it within surface layer or also above surface layer?

3. The authors mention LLJs, frequently observed in the ARM site, how does this effect the ML output at higher heights?

4. The idea of round-robin is fair for machine-learning based extrapolation, but only if the training has been done accounting for all atmospheric conditions that would be representative of other sites. As you know, ML models can only learn what is in their training dataset. Therefore, the round-robin type approaches come with a caveat that the search space of the variables expands to many of the common conditions (including external forcings specific to each site) observed in the atmosphere and at all the evaluated sites. This comment needs to be addressed in the paper with supporting evidence.

5. For the SNR filtering, not only precipitation, but fog is also prevalent at SGP and it diminishes the range considerably at lower heights. Therefore, an upper limit on SNR could be important to filter out any abnormalities in radial velocity data.

6. Line 94: Maybe I am picky, but the poor data quality is because those measurements fall within the lidar blind zone? The Blind zone is generally 2 times the range-gate size, which fits the heights. If yes, please mention that for clarity.

7. Equation 2: The temperature used was from the sonic or from the cup anemometer

for the fluxes? Sonic anemometer temperature measurements have significant biases and are not considered very accurate (Berg et al., 2017). This would cause errors in classifying stability or L.

8. How is atmospheric stability defined? Based on Richardson number of MO length? Please provide the thresholds or a reference from which you picked the thresholds for classifying stability for the MO type extrapolation.

9. MO length is not known to be valid for complex terrain (Fernando et al., 2015), therefore these parameters would not fit well for all types of terrain/sites. Therefore, a note about applicability of the chosen parameters to different conditions/terrains would be needed to address the universality of these parameters for such an approach.

10. The effect of external forcings at different ARM sites are not considered, which is important in this context of machine learning (comment #4 above). The wakes from wind turbines have major impact on the hub-height winds at some of these sites. Sites E37 and E39 are far away from turbines or wind farms, while C1 and E41 are relatively closer and have considerable impact on the winds at hub-height in certain predominant wind directions. Please see attached the wind directions and distance from wind turbines at each of these sites and something similar must be included in your analysis. Therefore, I would recommend you can either discard the below sectors from your analysis or test the accuracy in waked conditions.

11. How much of these chosen parameters (TKE, L, WS4, WS65) explain the variance in the RF model? What is the unbiased predictor importance estimates of the chosen variables?

12. Figure 7: Maybe some additional explanation is required on how the dependence is calculated. Its not very clear if its just a correlation type analysis or something else. Please provide more details here. Also, the extrapolated wind speeds (Y-axis) for all plots are not same and it's not very clear why.

Very Minor comment: The language is a bit colloquial for a journal and would urge the authors to take that into consideration for their revised manuscript. For example: Line 225: Maybe you can but I am not sure if it's formal to end a sentence with "are": please rephrase. Similar sentence structuring needs to be considered throughout the document.
* * *
| Lidar Location | Wind Direction Sectors | Approximate Distance of the nearest Turbine to the Lidar (m) | Common Turbine Height in that sector | Rotor Diameter | Type of Turbine | Built Year |
|---|---|---|---|---|---|---|
| C1 | 67 - 93 | 6700 | 90 | 116 | GE 2.5 MW | 2017 |
| | 112 - 196 | 3500 | 80 | 116 | GE 2.3 MW | 2017 |
| | 243 - 270 | 4600 | 80 | 82.5 | GE 1.68 MW | 2012 |
| E32 | 45 - 60 | 11500 | 80 | 108 | Siemens 2.3 MW | 2016 |
| E37 | -- | > 20000 | -- | -- | -- | -- |
| E39 | -- | > 20000 | -- | -- | -- | -- |
| E41 | 205-255 | 2500 | 87 | 126 | Vestas V126-3.3 | 2016 |
| | 295 - 15 | 5000 | 80 | 108 | Siemens 2.3 MW | 2015 |

**Fig. 1.** SGP ARM Doppler Lidars Wind Farm Distance

---

## Referee Comment (RC3) · Anonymous Referee #3 · 19 Feb 2020

The authors present a machine learning approach to vertical extrapolation of wind speeds compared to standard approaches. The research is quite robust, described in detail and well written. The conclusion that the machine learning approach, a random forest, can be extrapolated to other sites as shown by this round robin evaluation is an important scientific discovery.

There are a few areas that can be further described or clarified to make this an excellent paper.

1.) Page 5 line 108 - it is stated that precipitation periods were excluded from the analysis. Please explain why and what impact this has on the analysis.

2.) Page 5 Line 11 - it is stated that a 30-min average is used. Is there a reason why 30-min was chosen and would that averaging period affect the results? No further analysis is needed - just an explanation or including in the results discussion how the averaging period may impact the analysis.

3.) Page 7 Line 146 - please cite Pedregosa et al 2011 for Scikit-learn. http://www.jmlr.org/papers/v12/pedregosa11a.html

4.) Page 7 Lines 155-157 - the explanation of training, testing and cross-validation is not clear. Is the 5-fold cross validation performed on the 80% training data and the 20% testing data is held out for independent validation after the hyperparameters are chosen? Please describe so that it is clear the testing data was not used in the choosing of the hyperparamters.

5.) Figure 7 are the partial dependence plots for the random forest, which are an important aspect of the interpretability of the machine learning models. However, it is better to show both predictor importance and partial dependence plots so that the relative importance of each variable and it's associated partial dependence is known. Recommend adding in predictor importance plots or list to add value to the interpretability.

---

## Author Comment (AC2) · 5 Mar 2020

*In this document, the reviewer's comments are in black, the authors' responses are in red.*

The authors thank the reviewer for their thoughtful comments, which helped us improve the quality of our manuscript.

The paper presents a robust measurement campaign and analysis results. The thoroughness of the various analyses is commendable. The sensitivity analysis at the end of the paper is particularly insightful as it helps shed light on the reasons for the performance of the machine-learning (ML) approach used by the authors. Indeed, the risk when using ML is to blindly depend on a black box which may, or may not, provide reliable output. especially for new situations for which no data was included in the training set.

The following questions and comments are provided in the hope of enhancing the readability and overall reach of the paper:

0. One could argue that power law and log law are also machine learning approaches - even though they are simple regressions!
   Fair point! As the power law and log law are so well-known and broadly used in the wind energy community, we think referring to them as "conventional techniques" will be of easier understanding for the general reader of the paper.

1. In the intro, low-level jets (LLJ) are mentioned. Provide some more background as they are not ubiquitously present, nor relevant. Or specify that " in some regions ...".
   We have added "in some regions" in the introduction sentence.
   Also, we have extensively studied ML extrapolation for LLJ events in a companion conference paper which is currently in review. We have added the following sentence to the Results section, after the analysis of the ML extrapolation performance with height: "As an application of the performance of the random forest in predicting wind speed at higher heights, we present the case study of a LLJ in a companion paper (Bodini and Optis, in review)."

2. Provide a better presentation of the measurement campaign - notably, do not forget to add the missing paragraph which was posted: - Site description - Typical wind regime description - Lidar precision/accuracy/validation/testing discussion as the wind industry is still considering scanning lidars with a lot of caution. Or, provide discussion that high lidar accuracy is irrelevant in this context because ... - Provide an idea of the total number of data samples used. - Any data quality applied?
   We have included the paragraph that was missing in the first draft, and added details across Section 2 to include the suggestions of the reviewer. Section 2 now reads as follows:

 **2 Data: The Southern Great Plains (SGP) Atmospheric Observatory**

[revised manuscript text omitted]

3. The wind industry also uses by-sector and/or by-hour-of-day vertical extrapolation. These are targeting a couple of shortcomings the authors note, namely: stability and terrain complexity. It would be useful to add this in the discussion - or even better, in the analysis. We have added the following analysis to the results section:
"In addition, it is important to check whether the results of the performance comparison are affected by the time resolution at which the shear exponent α is calculated. Wind energy consultants apply a variety of methods to calculate shear (Brower, 2012): one could calculate shear values at each timestamp (as done in our analysis), or use a single average shear exponent, or consider various shear values based on bins of wind direction and/or time of day. To compare the time series-based shear calculation with its most different approach, we test the performance of the power law in extrapolating the average wind resource from 65 m AGL to 143 m AGL using only a single mean value for the shear exponent, calculated as the average of the α values at each considered timestamp. We find that the average extrapolated wind speed from the random forest approach still has a smaller error compared to the average extrapolated wind speed using the mean shear value, at all the considered sites (across-site MAE for random forest is 0.01 m s−1, for power law is 0.13 m s−1). Given the overall small MAE values found for both methods, we can also conclude that machine-learning-based extrapolation approaches are most beneficial for time series-based extrapolations, as deficiencies in conventional approaches tend to average out more when considering the long-term average results."

4. My understanding is that the authors optimized the hyper-parameters by making use of available target-height measurements. So what is the authors' suggestion to fine-tune these parameters in the absence of target-height measurements? Could we contemplate a

database of parameters for specific site conditions? Other? More generally, how their round-robin results could be leveraged, used on site?

We have added the following sentence to the Conclusions of the paper: "In real world applications, a machine learning algorithm could be trained on observations collected by a single lidar, and then used to extrapolate wind speed at nearby locations, where only much cheaper short meteorological masts would need to be installed".

5. Lines 192 and following: any particular reason comparison results for the specific use case under discussion were not more thoroughly reported?

We have added the following table to the Supplement (and added reference to it in this paragraph in the main paper) to support our description of the comparison between ML and power law performance when data at 91 m AGL are included in both methods:

Table 1: Percentage reduction in wind-speed extrapolation MAE from the random forest approach over the power law when wind shear is calculated using data at 4 m and 65 m AGL versus at 65 m and 91 m AGL. In the latter case, wind speed at 91 m AGL is included as input feature for the random forest model.

| | Training - testing site | | | | Average |
|---|---|---|---|---|---|
| Error reduction relative to POWER LAW | C1 | E37 | E39 | E41 | |
| Shear from 4 m and 65 m AGL | $-25\%$ | $-36\%$ | $-27\%$ | $-24\%$ | $-28\%$ |
| Shear from 65 m and 91 m AGL | $-15\%$ | $-22\%$ | $-16\%$ | $-15\%$ | $-17\%$ |

6. Personally, I find the last sections of the paper to be the most valuable ones! Without suggesting to re-write the whole paper, I submit the following ideas for author's consideration: - Put the emphasis on the fact that more physical parameters where included in a data-driven model, and their impact on model performance was investigated and fully understood (cf. sensitivities). - The model seems to out-perform standard models, even under round-robin conditions (which is indeed a better way of assessing the model). - The model could be used for a given site as follows (might need more thought to be put here ...)

In the Results section, we have added an extensive discussion of feature importance to further emphasize the importance of being able to understand and quantify the different input features used in the machine learning model:

**Table 5.** Predictor importance for the random forest used to extrapolate winds at 143 m AGL at site C1

| Predictor | Relative importance |
| --- | --- |
| WS 65 m | 68% |
| WS 4 m | 18% |
| time | 3% |
| L | 8% |
| TKE | 3% |

"The results of the analysis of the predictor performance are listed in Table 5. As already suggested by the partial dependence analysis, wind speed at 65 m AGL is the predictor with the largest importance in extrapolating wind speed at 143 m AGL. However, all the considered surface observations account for over 30% of the overall performance of the random forest. In particular, the addition of the Obukhov length to include direct atmospheric stability information in the algorithm has a not-negligible 8% importance."
We have also added the following sentences to the Conclusions:
"The benefit of including more physical parameters in a data-driven model clearly demonstrates its importance."
"In real world applications, a machine learning algorithm could be trained on observations collected by a single lidar, and then used to extrapolate wind speed at nearby locations, where only much cheaper short meteorological masts would need to be installed."
We have also rephrased the following sentence in the Conclusions to further emphasize that the round-robin validation still outperforms conventional techniques:
"Therefore, we have confirmed that the random-forest approach outperforms conventional techniques for wind-speed vertical extrapolation, even under a more robust round-robin validation, which we recommend to avoid overestimating the potential performance of machine-learning techniques, which could lead to underestimation of the uncertainty in wind speed estimates."

Thank you for having submitted a paper which makes a balanced and useful use of ML!
Thank you for taking the time to review our manuscript!

---

## Author Comment (AC3) · 5 Mar 2020

*In this document, the reviewer's comments are in black, the authors' responses are in red.*

The authors thank the reviewer for their thoughtful comments, which helped us improve the quality of our manuscript.

The article "The importance of round-robin validation when assessing machine- learning-based vertical extrapolation of wind speeds" by Bodini & Optis details a round robin approach to vertical extrapolation from the 4 ARM SGP Doppler lidars using a random forest algorithm. The paper is well written and the reviewer agrees the necessity of a round-robin type approach to assess the accuracy of machine learning algorithms and make it more universal. Below are some comments/questions which probe into some of the details of the paper and would improve the paper if addressed in the next version.

1. Line 61: Extrapolation is not only generally done up to Hub-height but through the rotor swept area. So, I am not sure I follow the author's argument here, that if hub- height winds are available extrapolation is unnecessary. The approach to go above hub-height can also be treated as a "Gap filling" approach for met-masts (when Lidars are moved around from one location to the other for a short period). Please clarify.
We agree with the reviewer that the approach should not be limited to hub-height wind speed extrapolation, but can rather be used to obtain wind resource at any height of interest for wind energy production. To make this clear, we have changed the wording "hub-height wind speed" to expressions such as "heights relevant for wind energy production" or "heights of the rotor swept area" throughout the manuscript.

2. For power law type extrapolations, measurements not only at the surface but at multiple heights is needed to estimate the dynamic power law exponent. So please define what you mean by near-surface in the paper? Is it within surface layer or also above surface layer?
We have rephrased the sentence in the introduction as "By contrast, conventional extrapolation approaches do not have nor require knowledge of hub-height wind speeds and therefore can generalize to any location where measurements are available at a single level near the surface (for the logarithmic law) or at two levels in the lower part of the boundary layer (for the power law).".

3. The authors mention LLJs, frequently observed in the ARM site, how does this effect the ML output at higher heights?
We have extensively studied ML extrapolation for LLJ events in a companion conference paper which is currently in review. We have added the following sentence to the Results section, after the analysis of the ML extrapolation performance with height: "As an application of the performance of the random forest in predicting wind speed at higher heights, we present the case study of a LLJ in a companion paper (Bodini and Optis, in review)." We will update the reference if the conference paper is reviewed before this manuscript is accepted for publication.

4. The idea of round-robin is fair for machine-learning based extrapolation, but only if the training has been done accounting for all atmospheric conditions that would be representative of other sites. As you know, ML models can only learn what is in their

training dataset. Therefore, the round-robin type approaches come with a caveat that the search space of the variables expands to many of the common conditions (including external forcings specific to each site) observed in the atmosphere and at all the evaluated sites. This comment needs to be addressed in the paper with supporting evidence.
See answer to comment 10.

5. For the SNR filtering, not only precipitation, but fog is also prevalent at SGP and it diminishes the range considerably at lower heights. Therefore, an upper limit on SNR could be important to filter out any abnormalities in radial velocity data.
We have re-done our analysis by setting an upper limit on SNR, chosen after inspecting the data. We have rephrased the sentence in Section 2.1 as: "We discard from the analysis measurements measurements with a signal-to-noise ratio lower than –21 dB or higher than +5 dB (to filter out fog events), along with periods of precipitation, as recorded by a disdrometer at the C1 site.".

6. Line 94: Maybe I am picky, but the poor data quality is because those measurements fall within the lidar blind zone? The Blind zone is generally 2 times the range-gate size, which fits the heights. If yes, please mention that for clarity.
We have rephrased the sentence as follows: "Data recorded at two lowest heights (13 and 39 m AGL) could not be used because of their poor quality, as they lie in the lidar blind zone."

7. Equation 2: The temperature used was from the sonic or from the cup anemometer for the fluxes? Sonic anemometer temperature measurements have significant biases and are not considered very accurate (Berg et al., 2017). This would cause errors in classifying stability or L.
We have discussed with the instrument mentor at ANL, who confirmed that the flux data provided on the ARM website have been linearly corrected to account for the instrument issues the reviewer is mentioning. On the other hand, for the average temperature data, which were not corrected, we have now switched to use data from the 2-m temperature and humidity probe as done in Berg et al. 2017. Section 2.2 now reads:

 **2.2 Surface Measurements**

Surface data were collected by sonic anemometers on flux measurement systems and temperature probes, which were deployed at each of the four considered sites. The sonic anemometer measured the three wind components at a 10-Hz resolution; processed data are available as 30-minute averages. We use wind speed at 4 m AGL, and turbulent kinetic energy (TKE) calculated from the variance of the three components of the wind flow as:

$$TKE = \frac{1}{2}(\sigma_u^2 + \sigma_v^2 + \sigma_w^2) \tag{1}$$

Also, at each site we calculate the Obukhov length, $L$, to quantify atmospheric stability:

$$L = -\frac{\overline{T_v} \cdot u_*^3}{k \cdot g \cdot \overline{w'T_v'}} \tag{2}$$

where $k = 0.4$ is the von Kármán constant; $g = 9.81 \text{ m s}^{-2}$ is the gravity acceleration; $T_v$ is the virtual temperature (K); $u_* = (\overline{u'w'}^2 + \overline{v'w'}^2)^{1/4}$ is the friction velocity ($\text{m s}^{-1}$); and $\overline{w'T_v'}$ is the kinematic virtual temperature flux ($\text{K m s}^{-1}$). A linear correction (Pekour, 2004) has been applied to the flux processing to account for sonic anemometer deficiencies in measuring temperature at sites E37, E39, and E41. For the same reason, at these sites, we use $\overline{T_v}$ from temperature and humidity probes at 2 m AGL. Reynolds decomposition for turbulent fluxes has been applied using a 30-minute averaging period, as commonly chosen for boundary-layer processes (De Franceschi and Zardi, 2003; Babić et al., 2012). We consider stable conditions for $L > 0\text{m}$, and unstable conditions for $L < 0\text{m}$. Data have been quality-controlled, and precipitation periods were excluded from the analysis to discard inaccurate measurements (Zhang et al., 2016).

8. How is atmospheric stability defined? Based on Richardson number of MO length? Please provide the thresholds or a reference from which you picked the thresholds for classifying stability for the MO type extrapolation.
   We have added the following sentence in Section 2.2: "We consider stable conditions for L > 0 m, and unstable conditions for L < 0 m.".

9. MO length is not known to be valid for complex terrain (Fernando et al., 2015), therefore these parameters would not fit well for all types of terrain/sites. Therefore, a note about applicability of the chosen parameters to different conditions/terrains would be needed to address the universality of these parameters for such an approach.
   We have added the following sentence to Section 3.3: "We note that when similar techniques are applied to more complex sites, the Obukhov length might not be well-suited to capture atmospheric stability in complex terrain (Fernando et al., 2015), and therefore an accurate choice of the input variables as a function of the specific topography is recommended.".

10. The effect of external forcings at different ARM sites are not considered, which is important in this context of machine learning (comment #4 above). The wakes from wind turbines have major impact on the hub-height winds at some of these sites. Sites E37 and E39 are far away from turbines or wind farms, while C1 and E41 are relatively closer and have considerable impact on the winds at hub-height in certain predominant wind directions. Please see attached the wind directions and distance from wind turbines at each of these sites and something similar must be included in your analysis. Therefore, I would recommend you can either discard the below sectors from your analysis or test the accuracy in waked conditions.

We agree with the reviewer that different forcings experienced at different sites have an importance when assessing the round-robin validation of the proposed machine learning method, and that explicit emphasis on this caveat should be included in the analysis. For the specific comment about the impact of wind farms, we have extensively studied this topic in the aforementioned companion conference paper.

We have added the following discussion paragraph to the Results section to make all these thoughts explicit to the reader:

"Moreover, we can expect the performance comparison to be influenced not only by the pure separation between training and testing sites, but also by the different forcings that each specific site experiences. Notably, Bodini and Optis (in review) compared the extrapolation performance of the proposed random forest approach before and after a wind farm was built in the vicinity of site C1, and found an increase in MAE up to 10% if waked data are not included in the training set. Therefore, to fully exploit the performance of the proposed machine learning approach in extrapolating the wind resource at sites different from the training one it is essential to build a training set of observations which can encompass the specific atmospheric conditions representative of the desired testing site."

11. How much of these chosen parameters (TKE, L, WS4, WS65) explain the variance in the RF model? What is the unbiased predictor importance estimates of the chosen variables?

We have added the predictor importance analysis as suggested by the reviewer. The following paragraph has been added:

"The results of the analysis of the predictor performance are listed in Table 5. As already suggested by the partial dependence analysis, wind speed at 65 m AGL is the predictor with the largest importance in extrapolating wind speed at 143 m AGL. However, all the considered surface observations account for over 30% of the overall performance of the random forest. In particular, the addition of the Obukhov length to include direct atmospheric stability information in the algorithm has a not-negligible 8% importance."

The following table has also been included in the manuscript:

**Table 5.** Predictor importance for the random forest used to extrapolate winds at 143 m AGL at site C1

| Predictor | Relative importance |
|---|---|
| WS 65 m | 68% |
| WS 4 m | 18% |
| time | 3% |
| L | 8% |
| TKE | 3% |

12. Figure 7: Maybe some additional explanation is required on how the dependence is calculated. It's not very clear if it's just a correlation type analysis or something else. Please provide more details here. Also, the extrapolated wind speeds (Y-axis) for all plots are not same and it's not very clear why.

We have improved our introduction to Figure 7 in the results section, and added a reference where more information on partial dependence analysis can be found. The paragraph now reads: "Figure 7 shows the partial dependence plots, which show the marginal effect of

each input feature on the predicted extrapolated wind speed (Friedman, 2001). We note that the values on the y-axes have not been normalized, so that large ranges indicate strong dependence of extrapolated wind speed on the feature, whereas small ranges show weaker dependence.".

Very Minor comment: The language is a bit colloquial for a journal and would urge the authors to take that into consideration for their revised manuscript. For example: Line 225: Maybe you can but I am not sure if it's formal to end a sentence with "are": please rephrase. Similar sentence structuring needs to be considered throughout the document.

We have rephrased the sentence as "Distributions of the input features are also shown, which help distinguish densely populated regions, with strong statistical relationships, and sparsely populated regions, with weaker statistical relationships." The whole manuscript has undergone editorial review by a professional native English-speaking editor.

---

## Author Comment (AC4) · 5 Mar 2020

*In this document, the reviewer's comments are in black, the authors' responses are in red.*

The authors thank the reviewer for their thoughtful comments, which helped us improve the quality of our manuscript.

The authors present a machine learning approach to vertical extrapolation of wind speeds compared to standard approaches. The research is quite robust, described in detail and well written. The conclusion that the machine learning approach, a random forest, can be extrapolated to other sites as shown by this round robin evaluation is an important scientific discovery.

There are a few areas that can be further described or clarified to make this an excellent paper.

1.) Page 5 line 108 - it is stated that precipitation periods were excluded from the analysis. Please explain why and what impact this has on the analysis.
We have rephrased the sentence and added a reference to a study on the impact of precipitation on the accuracy of sonic anemometer data. It now reads: "precipitation periods were excluded from the analysis to discard inaccurate measurements (Zhang et al. 2016)."

2.) Page 5 Line 11 - it is stated that a 30-min average is used. Is there a reason why 30-min was chosen and would that averaging period affect the results? No further analysis is needed - just an explanation or including in the results discussion how the averaging period may impact the analysis.
Because of the wrong line number listed, we could not determine whether the reviewer is referring to the 30-minute average period used for the lidar and sonic anemometer data, or for the 30-minute average period used for the Reynolds decomposition to calculate Obukhov length. In any case:
1) For the 30-minute average period used as resolution for the main data used in the analysis, the choice was due to the fact the sonic anemometer data were only publicly available at that time resolution. We have added the following sentence to Section 2.2 "processed data are available as 30-minute averages". We have also added the following comment to the beginning of Section 3: "We acknowledge that the resolution of the data used will have an impact on the magnitude of the error values shown in the analysis (as observations at a higher time resolution would likely cause larger extrapolation errors). However, we do not expect the relative comparison between the different extrapolation techniques and the analysis of the predictor importance to be strongly affected by the resolution of the input features used."
2) For the 30-minute average period used for the Reynolds decomposition, as stated in the paragraph (with appropriate references listed), 30-minute is the most common averaging period used to calculate fluxes for boundary layer processes, as it is considered to be shorter than the period of large-scale fluctuations, but longer than the period of short-term turbulence fluctuations, following considerations related to the spectral gap (Van Der Hoven, 1957).

3.) Page 7 Line 146 - please cite Pedregosa et al 2011 for Scikit-learn. http://www.jmlr.org/papers/v12/pedregosa11a.html

We have added the suggested reference.

4.) Page 7 Lines 155-157 - the explanation of training, testing and cross-validation is not clear. Is the 5-fold cross validation performed on the 80% training data and the 20% testing data is held out for independent validation after the hyperparameters are chosen? Please describe so that it is clear the testing data was not used in the choosing of the hyperparameters.
We have rephrased the paragraph as: "We use a five-fold cross validation to evaluate different combinations of the hyperparameters, with 30 sets randomly sampled at each site. We use 80% of the data in the cross-validation, while the remaining 20% (selected without shuffling the original data set to avoid unfair predicting performance improvement because of auto-correlation in the data) is held out for independent testing."

5.) Figure 7 are the partial dependence plots for the random forest, which are an important aspect of the interpretability of the machine learning models. However, it is better to show both predictor importance and partial dependence plots so that the relative importance of each variable and its associated partial dependence is known. Recommend adding in predictor importance plots or list to add value to the interpretability.
We have added the predictor performance analysis as suggested by the reviewer. The following paragraph has been added:
"The results of the analysis of the predictor performance are listed in Table 5. As already suggested by the partial dependence analysis, wind speed at 65 m AGL is the predictor with the largest importance in extrapolating wind speed at 143 m AGL. However, all the considered surface observations account for over 30% of the overall performance of the random forest. In particular, the addition of the Obukhov length to include direct atmospheric stability information in the algorithm has a not-negligible 8% importance."
The following table has also been included in the manuscript:

**Table 5.** Predictor importance for the random forest used to extrapolate winds at 143 m AGL at site C1

| Predictor | Relative importance |
| --- | --- |
| WS 65 m | 68% |
| WS 4 m | 18% |
| time | 3% |
| L | 8% |
| TKE | 3% |